# Uncertainty Weighted Offline Reinforcement Learning

## Abstract

Offline Reinforcement Learning promises to learn effective policies from previously-collected, static datasets without the need for exploration. However, existing Q-learning and actor-critic based off-policy RL algorithms fail when bootstrapping from out-of-distribution (OOD) actions or states. We hypothesize that a key missing ingredient from the existing methods is a proper treatment of uncertainty in the offline setting. We propose Uncertainty Weighted Actor-Critic (UWAC), an algorithm that models the epistemic uncertainty to detect OOD state-action pairs and down-weights their contribution in the training objectives accordingly. Implementation-wise, we adopt a practical and effective dropout-based uncertainty estimation method that introduces very little overhead over existing RL algorithms. Empirically, we observe that UWAC substantially improves model stability during training. In addition, UWAC out-performs existing offline RL methods on a variety of competitive tasks, and achieves significant performance gains over the state-of-the-art baseline on datasets with sparse demonstrations collected from human experts.

## 1 Introduction

Deep reinforcement learning (RL) has seen a surge of interest over the recent years. It has achieved remarkable success in simulated tasks (Silver et al., 2017; Schulman et al., 2017; Haarnoja et al., 2018), where the cost of data collection is low. However, one of the drawbacks of RL is its difficulty of learning from prior experiences. Therefore, the application of RL to unstructured real-world tasks is still in its primal stages, due to the high cost of active data collection. It is thus crucial to make full use of previously collected datasets whenever large scale online RL is infeasible.

Offline batch RL algorithms offer a promising direction to leveraging prior experience (Lange et al., 2012). However, most prior off-policy RL algorithms (Haarnoja et al., 2018; Munos et al., 2016; Kalashnikov et al., 2018; Espeholt et al., 2018; Peng et al., 2019) fail on offline datasets, even on expert demonstrations (Fu et al., 2020). The sensitivity to the training data distribution is a well known issue in practical offline RL algorithms (Fujimoto et al., 2019; Kumar et al., 2019; 2020; Peng et al., 2019; Yu et al., 2020). A large portion of this problem is attributed to actions or states not being covered within the training set distribution. Since the value estimate on out-of-distribution (OOD) actions or states can be arbitrary, OOD value or reward estimates can incur destructive estimation errors that propagates through the Bellman loss and destabilizes training. Prior attempts try to avoid OOD actions or states by imposing strong constraints or penalties that force the actor distribution to stay within the training data (Kumar et al., 2019; 2020; Fujimoto et al., 2019; Laroche et al., 2019). While such approaches achieve some degree of experimental success, they suffer from the loss of generalization ability of the $Q$ function. For example, a state-action pair that does not appear in the training set can still lie within the training set distribution, but policies trained with strong penalties will avoid the unseen states regardless of whether the $Q$ function can produce an accurate estimate of the state-action value. Therefore, strong penalty based solutions often promote a pessimistic and sub-optimal policy. In the extreme case, e.g., in certain benchmarking environments with human demonstrations, the best performing offline algorithms only achieve the same performance as a random agent (Fu et al., 2020), which demonstrates the need of robust offline RL algorithms.

In this paper, we hypothesize that a key aspect of a robust offline RL algorithm is a proper estimation and usage of uncertainty. On the one hand, one should be able to reliably assign an uncertainty score

to any state-action pair; on the other hand, there should be a mechanism that utilizes the estimated uncertainty to prevent the model from learning from data points that induce high uncertainty scores.

The first problem relates closely to OOD sample detection, which has been extensively studied in the Bayesian deep learning community. (Gal & Ghahramani, 2016a; Gal, 2016; Osawa et al., 2019), often measured by the epistemic uncertainty of the model. We adopt the dropout based approach Gal & Ghahramani (2016a), due to its simplicity and empirical success in practice. For the second problem, we provide an intuitive modification to the Bellman updates in actor-critic based algorithms. Our proposed Uncertainty Weighted Actor Critic (UWAC) is to simply down weigh the contribution of target state and action pairs with high uncertainty. By doing so, we prevent the $Q$ function from learning from overly optimistic targets that lie far away from training data distribution (high uncertainty).

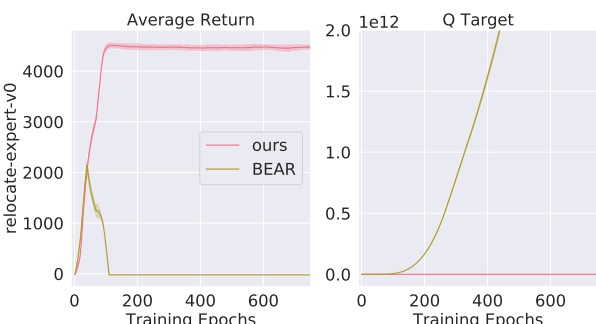

Figure 1: **Left.** Plot of average return v.s. training epochs of our proposed method (red) v.s. baseline (brown) (Kumar et al., 2019) on the relocate-expert offline dataset. **Right.** Corresponding plot of Q-Target values v.s. training epochs. Our proposed method achieves much higher average return, with better training stability, and more controlled Q-values.

Empirically, we first verified the effectiveness of dropout uncertainty estimation at detecting OOD samples. We show that the uncertainty estimation makes intuitive sense in a simple environment. With the uncertainty based down weighting scheme, our method significantly improves the training stability over our chosen baseline (Kumar et al., 2019), and achieves state-of-the-art performance in a variety of standard benchmarking tasks for offline RL.

Overall, our contribution can be summarized as follows: 1) We propose a simple and efficient technique (UWAC) to counter the effect of OOD samples with no additional loss terms or models. 2) We experimentally demonstrate the effectiveness of dropout uncertainty estimation for RL. 3) UWAC offers a novel way for stabilizing offline RL. 4) UWAC achieves SOTA performance on common offline RL benchmarks, and obtains significant performance gain on narrow human demonstrations.

## 2 RELATED WORK

In this work, we consider offline batch reinforcement learning (RL) under static datasets. Offline RL algorithms are especially prone to errors from inadequate coverage of the training set distribution, distributional shifts during actor critic training, and the variance induced by deep neural networks. Such error have been extensively studied as "error propagation" in approximate dynamic programming (ADP) (Bertsekas & Tsitsiklis, 1996; Farahmand et al., 2010; Munos, 2003; Scherrer et al., 2015). Scherrer et al. (2015) obtains a bound on the point-wise Bellman error of approximate modified policy iteration (AMPI) with respect to the supremum of the error in function approximation for an arbitrary iteration. We adopt the theoretical tools from (Kumar et al., 2019) and study the accumulation and propagation of Bellman errors under the offline setting.

One of the most significant problems associated with off-policy and offline RL is the bootstrapping error (Kumar et al., 2019): When training encounters an action or state unseen within the training set, the critic value estimate on out-of-distribution (OOD) samples can be arbitrary and incur an error that destabilizes convergence on all other states (Kumar et al., 2019; Fujimoto et al., 2019) through the Bellman backup. Yu et al. (2020) trains a model of the environment that captures the epistemic uncertainty. The uncertainty estimate is used to penalize reward estimation for uncertain states and actions, promoting a pessimistic policy against OOD actions and states. The main drawback of such a model based approach is the unnecessary introduction of a model of the environment – it is often very hard to train a good model. On the other hand, model-free approaches either train an agent pessimistic to OOD states and actions (Wu et al., 2019; Kumar et al., 2020) or constrain the actor distribution to the training set action distribution (Fujimoto et al., 2019; Kumar et al., 2019; Wu et al., 2019; Jaques et al., 2019; Fox et al., 2015; Laroche et al., 2019). However, the pessimistic assumption that all unseen states or actions are bad may lead to a sub-optimal agent and

greatly reduce generalization to online fine-tuning (Nair et al., 2020). Distributional constraints, in addition, rely on approximations since the actor distribution is often implicit. Such approximations cause practical training instability that we will study in detail in section 5.4.

We propose a model-free actor-critic method that down-weighs the Bellman loss term by inverse uncertainty of the critic target. Uncertainty estimation has been implemented in model-free RL for safety and risk estimation (Clements et al., 2019; Hoel et al., 2020) or exploration (Gal & Ghahramani, 2016a; Lines & Van Der Wilk), through ensembling (Hoel et al., 2020) or distributional RL (Dabney et al., 2018; Clements et al., 2019). However, distributional RL works best on discrete action spaces (Dabney et al., 2018) and require additional distributional assumptions when extended to continuous action spaces (Clements et al., 2019). Our approach estimates uncertainty through Monte Carlo dropout (MC-dropout) (Srivastava et al., 2014). MC-dropout uncertainty estimation is a simple method with minimal overhead and has been thoroughly studied in many traditional supervised learning tasks in deep learning (Gal & Ghahramani, 2016a; Hron et al., 2018; Kingma et al., 2015; Gal & Ghahramani, 2016b). Moreover, we observe experimentally that MC-dropout uncertainty estimation behaves similarly to explicit ensemble models where the prediction is the mean of the ensembles, while being much simpler (Lakshminarayanan et al., 2017; Srivastava et al., 2014).

The most relevant to our work are MOPO (Yu et al., 2020), BEAR (Kumar et al., 2019), and CQL (Kumar et al., 2020). MOPO approaches offline RL from a different model-based paradigm, and obtains competitive results on some tasks with wide data distribution. However, due to the model-based nature, MOPO achieves limited performance on most other benchmarks due to the performance of the model being limited by the data distribution. On the other hand, BEAR and CQL both use actor-critic and do not suffer from the above problem. We use BEAR (discussed in section 3.2) as our baseline algorithm and achieve significant performance gain through dropout uncertainty weighted backups. CQL avoids OOD states/actions through direct $Q$ value penalty on actions that leads to OOD unseen states within the training set. However the penalty proposed by CQL 1) risks hurting $Q$ estimates for (action, state) pairs that are not OOD, since samples not seen within the dataset can still lie within the true dataset distribution; 2) limits the policy to be pessimistic, which may be hard to fine-tune once on-policy data becomes available. Additionally our method is not limited to BEAR and can apply to other actor-critic methods like CQL. We leave such exploration to future works.

## 3 PRELIMINARIES

### 3.1 NOTATIONS

Following Kumar et al. (2019), we represent the environment as a Markov decision process (MDP) comprising of a 6-tuple $(\mathcal{S}, \mathcal{A}, P, R, \rho_0, \gamma)$, where $\mathcal{S}$ is the state space, $\mathcal{A}$ is the action space, $P(s'|s, a)$ is the transition probability distribution, $\rho_0$ is the initial state distribution, $R : \mathcal{S} \times \mathcal{A} \to \mathbb{R}$ is the reward function, and $\gamma \in (0, 1]$ is the discount factor. Our goal is to find a policy $\pi(s|a)$ from the set of policy functions $\Pi$ to maximize the expected cumulative discounted reward.

Standard Q-learning learns an optimal state-action value function $Q^*(s, a)$, representing the expected cumulative discounted reward starting from $s$ with action $a$ and then acting optimally thereafter. Q-learning is trained on the Bellman equation defined as follows with the Bellman optimal operator $\mathcal{T}$ defined by:

$$\mathcal{T}Q(s, a) := R(s, a) + \gamma \mathbb{E}_{P(s'|s,a)} \left[ \max_{a'} Q(s', a') \right] \tag{1}$$

In practice, the critic ($Q$ function) is updated through dynamic programming, by projecting the target $Q$ estimate ($\mathcal{T}Q$) into $Q$ (i.e. minimizing Bellman Squared Error $\mathbb{E}\left[(Q - \mathcal{T}Q)^2\right]$). Since $\max_{a'} Q(s', a')$ in generally intractable in continuous action spaces, an actor ($\pi_\theta$) function is learned to maximize the critic function ($\pi_\theta(s) := \arg\max_a Q(s, a)$) (Haarnoja et al., 2018; Fujimoto et al., 2018; Sutton & Barto, 2018).

In the context of offline reinforcement learning, naively performing $\max_{a'} Q(s', a')$ in equation 1 may result in an $a'$ unseen within the training dataset (OOD), and resulting in a $Q$ estimate with very large error that can propagate through the Bellman bootstrapping and destabilize training on other states (Kumar et al., 2019).

## 3.2 BASELINE ALGORITHM

We use BEAR (Kumar et al., 2019) as our baseline algorithm. BEAR restricts the set of policy functions ($\Pi^\epsilon$) to output actions that lies in the support of the training distribution:

$$\pi(\cdot|s) := \arg \max_{\pi' \in \Pi^\epsilon} \mathbb{E}_{a \sim \pi'(\cdot|s)} [Q(s,a)] \tag{2}$$

Since the true support of $\pi \in \Pi^\epsilon$ is intractable. Kumar et al. (2019) instead relies on an approximate support constraint through optimizing sampled maximum mean discrepancy (MMD) (Gretton et al., 2012) between the training action distribution and the policy distribution.

However, this constraint eliminates the possibility of the $Q$ function to learn to generalize to state-action pairs beyond the training dataset and therefore limits the agent's performance and generalization. Moreover, the justification behind the sampled MMD approximation as support constraints is largely based on empirical evidence, and we observe numeric instability caused by discrepancies between $Q$ estimates and average returns on some narrower offline datasets (Figure 1). Such observations also correspond to Kumar et al. (2019)'s description in section 7.

## 4 UNCERTAINTY WEIGHTED OFFLINE RL

Our approach (UWAC) is motivated by connecting offline RL with the well-established Bayesian uncertainty estimation methods. This connection enables UWAC to "identify" and "ignore" OOD training samples, with no additional models or constraints.

### 4.1 UNCERTAINTY ESTIMATION THROUGH DROPOUT

Let $X$ capture all the state-action pairs in the training set: $X = (s,a)$, and $Y$ capture the true $Q$ value of the states. We follow a Bayesian formulation for the $Q$ function in RL parameterized by $\theta$, and maximize $p(\theta|X,Y) = p(Y|X,\theta)p(\theta)/p(Y|X)$ as our objective. Since $p(Y|X)$ is generally intractable, we approximate the inference process through dropout variational inference (Gal & Ghahramani, 2016a), by training with dropout before every weight layer, and also performing dropout at test time (referred to as Monte Carlo dropout). The epistemic uncertainty is captured by the approximate predictive variance with respect to the estimated $\hat{Q}$ for $T$ stochastic forward passes

$$Var[Q(s,a)] \approx \sigma^2 + \frac{1}{T} \sum_{t=1}^{T} \hat{Q}_t(s,a)^\top \hat{Q}_t(s,a) - E[\hat{Q}(s,a)]^\top E[\hat{Q}(s,a)]$$

with $\sigma^2$ corresponding to the inherent noise in the data, the second term corresponding to how much the model is uncertain about its predictions, and $E[\hat{Q}(s,a)]$ the predictive mean. We therefore use the second−third term to capture model uncertainty for OOD sample detection.

Overall, instead of training a $Q$ function on the policy $\pi$, we define an uncertainty-weighted policy distribution $\pi'$ with respect to the original policy distribution $\pi(\cdot|s)$ and normalization factor $Z(s)$

$$\pi'(a|s) = \frac{\beta}{Var[Q(s,a)]} \pi(a|s)/Z(s); \qquad Z(s) = \int_a \frac{\beta}{Var[Q(s,a)]} \pi(a|s) da \tag{3}$$

We show in the appendix A.1 that optimizing $\pi'$ results in theoretically better convergence properties against OOD training samples.

### 4.2 UNCERTAINTY WEIGHTED ACTOR-CRITIC

Instead of training the $Q$ function on Equation 1, we train $Q_\theta$ on $\pi'$. For clarity, we denote the TD $Q$-target as in (Mnih et al., 2013; Kumar et al., 2019) by $Q_{\theta'}$.

$$
\begin{aligned}
\mathcal{L}(Q_\theta) &= \mathbb{E}_{(s'|s,a) \sim \mathcal{D}} \mathbb{E}_{a' \sim \pi'(\cdot|s')} \left[ (Q_\theta(s,a) - (R(s,a) + \gamma Q_{\theta'}(s',a')))^2 \right] \\
&= \mathbb{E}_{(s'|s,a) \sim \mathcal{D}} \mathbb{E}_{a' \sim \pi(\cdot|s')} \left[ \frac{\beta}{Var[Q_{\theta'}(s',a')]} (Q_\theta(s,a) - (R(s,a) + \gamma Q_{\theta'}(s',a')))^2 \right]
\end{aligned} \tag{4}
$$

We absorb the normalization factor $Z$ into $\beta$. The resulting training loss down-weighs the Bellman loss for the $Q$ function by inverse the uncertainty of the $Q$-target ($Q_{\theta'}(s', a')$) that does track gradient. This directly reduces the effect that OOD backups has on the overall training process.

Similarly, we optimize the actor $\pi$ using samples from $\pi'$. Substituting $\pi(\cdot|s)$ by $\pi'(\cdot|s)$ in equation 2, we arrive at the following actor loss

$$\mathcal{L}(\pi) = -\mathbb{E}_{a\sim\pi'(\cdot|s)}\left[Q_\theta(s,a)\right] = -\mathbb{E}_{a\sim\pi(\cdot|s)}\left[\frac{\beta}{Var\left[Q_\theta(s,a)\right]}Q_\theta(s,a)\right] \tag{5}$$

The resulting actor loss intuitively decreases the probability of maximizing the Q function on OOD samples, further discouraging the vicious cycle of Q function explosion. Such loss further stabilizes $Q$ function estimations without constraints on the actor function distribution.

Algorithm 1 summarizes the proposed training curriculum, mostly the same as in the baseline (Kumar et al., 2019). Note that we do not propagate gradient through the uncertainty ($Var(y(s, a))$)

---

**Algorithm 1** Pseudo code for UWAC, differences from (Kumar et al., 2019) are colored

---

**Input:** Dataset $\mathcal{D}$, target network update rate $\tau$, mini-batch size $N$, sampled actions for MMD ($n = 10$), sample numbers stochastic forward passes ($T = 100$), hyper-parameters: $\lambda$, $\alpha$, $\beta$

1: Initialize Q networks $\{Q_{\theta_1}, Q_{\theta_2}\}$ with MC Dropout. Initialize actor $\pi_\phi$, target networks $\{Q_{\theta'_1}, Q_{\theta'_2}\}$ and a target actor $\pi_{\phi'}$, with $\phi' \leftarrow \phi, \theta'_{1,2} \leftarrow \theta_{1,2}$
2: **for** $t \leftarrow 1$ to $N$ **do**
3:     Sample mini-batch of transitions $(s, a, r, s') \sim \mathcal{D}$
4:     **Q-update:**
5:     Sample $p$ action samples, $\{a_i \sim \pi_{\phi'}(\cdot|s')\}_{i=1}^p$
6:     $y(s, a) := \max_{a_i}\left[\lambda \min(Q_{\theta'_1}(s', a_i), Q_{\theta'_2}(s', a_i)) + (1-\lambda)\max(Q_{\theta'_1}(s', a_i), Q_{\theta'_2}(s', a_i))\right]$

7:     Calculate variance of the $y(s, a)$ through variance of $T$ stochastic samples from $Q_{\theta'_1}, Q_{\theta'_2}$
8:     Perform one step of SGD to minimize $\mathcal{L}(Q_{\theta_{1,2}}) = \frac{\beta}{Var[y(s,a)]}(Q_{\theta_{1,2}}(s, a) - (r + \gamma y(s, a)))^2$

9:     **Policy-update:**
10:    Sample actions $\{a_i \sim \pi_{\phi'}(\cdot|s')\}_{i=1}^m$ and $\{a_j \sim \mathcal{D}\}_{i=1}^n$
11:    Update $\phi, a$ according to equation 5 with MMD penalty with weight $\alpha$ as in section 3.2
12:    **Update Target Networks:** $\theta'_{1,2} \leftarrow \tau\theta_{1,2}; \phi'_i \leftarrow \tau\phi_i$
13: **end for**

---

## 5 EXPERIMENTAL RESULTS

Our experiments are structured as follows: In section 5.1, we validate and visualize the effectiveness of dropout uncertainty estimation in RL. In section 5.2 we present competitive benchmarking results on the widely-used D4RL MuJoCo walkers dataset. We then experiment with the more complex Adroit hand manipulation environment in section 5.3, and analyze the training stability and the effectiveness against OOD samples by examining the Q target functions in section 5.4. We report the implementation details[1] and ablation studies in appendix A.2.

### 5.1 DROPOUT UNCERTAINTY ESTIMATION FOR REINFORCEMENT LEARNING

For the ease of 2D-visualization, we investigate our dropout uncertainty framework on the OpenAI gym LunarLander-v2 environment. The LunarLander-v2 environment features a lunar lander agent trying to land at a goal location in a 2D world (between two yellow flags) with 4 actions {do nothing, fire left engine, fire downward engine, fire right engine}.

We generate the expert offline dataset from the final replay buffer (size 100,000) of a fully trained expert AWR (Peng et al., 2019) agent with average reward 270. Note that the state-action distribution has a relatively complete coverage over the observation space (Fig. 2).

To simulate the scenario in most offline datasets, where the agent encounters lots of out-of-distribution states and actions, we create two skewed datasets by removing all observations from the

---

[1]We will release our code at github.com/anonymous

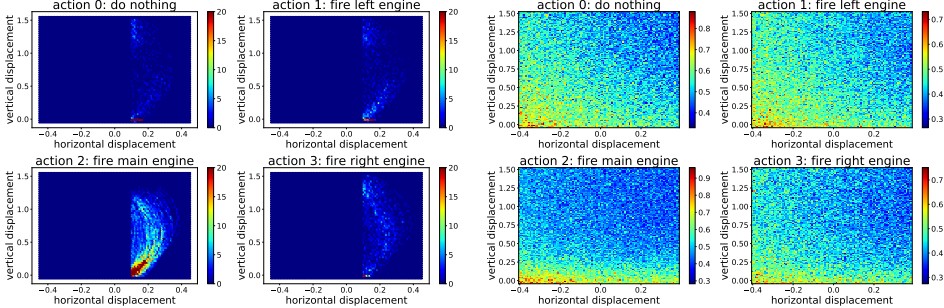

Figure 2: **Expert Trajectory Visualization.** 2D heat maps of the expert's action distribution with respect to horizontal/vertical displacement from the goal location. Warmer locations represent more observations.

Figure 3: **Left.** The training set with horizontal displacements ($< 0.1$) removed. This makes all states on the left OOD. **Right.** Our model estimates higher uncertainty (brighter color) on the left and lower uncertainty (colder color) on the right.

upper-half or the leftmost-half according to displacement from objective. We visualize the clipped datasets distribution together with the estimated Q function uncertainty in Figure 3,6. Our proposed framework reports higher uncertainty estimates at locations where the observations are sparse, especially where the observations are removed (OOD states). The results demonstrate the effectiveness of our proposed method at estimating the epistemic uncertainty of the Q function.

## 5.2 Performance on standard benchmarking datasets for offline RL

We evaluate our method on the MuJoCo datasets in the D4RL benchmarks (Fu et al., 2020), including three environments (halfcheetah, hopper, and walker2d) and five dataset types (random, medium, medium-replay, medium-expert, expert), yielding a total of 15 problem settings. The datasets in this benchmark have been generated as follows: **random**: roll out a randomly initialized policy for 1M steps. **expert**: 1M samples from a policy trained to completion with SAC. **medium**: 1M samples from a policy trained to approximately 1/3 the performance of the expert. **medium-replay**: replay buffer of a policy trained up to the performance of the medium agent. **medium-expert**: 50-50 split of medium and expert data.

Results are shown in Table 1. Our method is the strongest by a significant margin on all the medium-expert datasets and most of the medium-expert datasets, and also achieves good performance on all of the random and medium datasets, where the datasets lack state/action diversity. Our approach performs less well on the medium-replay datasets compared to model based method (MOPO) because model-based methods typically perform well on datasets with diverse state/action.

## 5.3 Performance on Adroit hand dataset with human demonstrations

We then experiment with a more complex robotic hand manipulation dataset. The Adroit dataset in the D4RL benchmarks (Rajeswaran et al., 2017) involves controlling a 24-DoF simulated hand to perform 4 tasks including hammering a nail, opening a door, twirling a pen, and picking/moving a ball. This dataset is particularly hard for previous state-of-the-art works in that it contains of narrow human demonstrations on a high-dimensional robotic manipulation task.

The dataset contains three types of datasets for each task. **human**: a small amount of demonstration data from a human; **expert**: a large amount of expert data from a fine-tuned RL policy; **cloned**: the third dataset is generated by imitating the human data, running the policy, and mixing data at a 50-50 ratio with the demonstrations. It is worth noting that mixing (for cloned) is performed because the cloned policies themselves do not successfully complete the task, making the dataset otherwise difficult to learn from (Fu et al., 2020).

Table 1: Normalized Average Returns on the D4RL MuJoCo Gym dataset according to (Fu et al., 2020). We report the average over 5 random seeds (± standard deviation). Fu et al. (2020); Kumar et al. (2020) do not report standard deviation. We omit cREM, BRAC-p, aDICE, and SAC-off because they do not obtain performance meaningful for comparison. We **bold** the highest mean.

| Task Name | UWAC (OURS) | MOPO | BEAR | BRACv | AWR | BCQ | BC | CQL |
|---|---|---|---|---|---|---|---|---|
| halfcheetah-random | $14.5 \pm 3.3$ | $31.9 \pm 3.9$ | 25.1 | 31.2 | 2.5 | 2.2 | 2.1 | **35.4** |
| walker2d-random | $\mathbf{15.5} \pm 11.7$ | $13.3 \pm 6.0$ | 7.3 | 1.9 | 1.5 | 4.9 | 1.6 | 7 |
| hopper-random | $\mathbf{22.4} \pm 12.1$ | $13.0 \pm 8.1$ | 11.4 | 12.2 | 10.2 | 10.6 | 9.8 | 10.8 |
| halfcheetah-medium | $\mathbf{46.5} \pm 2.5$ | $40.2 \pm 22.7$ | 41.7 | 46.3 | 37.4 | 40.7 | 36.1 | 44.4 |
| walker2d-medium | $57.5 \pm 7.8$ | $26.5 \pm 3.3$ | 59.1 | **81.1** | 17.4 | 53.1 | 6.6 | 79.2 |
| hopper-medium | $\mathbf{88.9} \pm 12.2$ | $14.0 \pm 7.6$ | 52.1 | 31.1 | 35.9 | 54.5 | 29.0 | 58 |
| halfcheetah-med-replay | $46.8 \pm 3.0$ | $\mathbf{54.0} \pm 12.6$ | 38.6 | 47.7 | 40.3 | 38.2 | 38.4 | 46.2 |
| walker2d-med-replay | $27.0 \pm 6.3$ | $\mathbf{92.5} \pm 30.4$ | 19.2 | 0.9 | 15.5 | 15.0 | 11.3 | 26.7 |
| hopper-med-replay | $39.4 \pm 6.1$ | $42.7 \pm 12.7$ | 33.7 | 0.6 | 28.4 | 33.1 | 11.8 | **48.6** |
| halfcheetah-med-expert | $\mathbf{127.4} \pm 3.7$ | $57.9 \pm 9.5$ | 53.4 | 41.9 | 52.7 | 64.7 | 35.8 | 62.4 |
| walker2d-med-expert | $\mathbf{99.7} \pm 12.2$ | $51.7 \pm 34.5$ | 40.1 | 81.6 | 53.8 | 57.5 | 6.4 | 98.7 |
| hopper-med-expert | $\mathbf{134.7} \pm 21.2$ | $55.0 \pm 3.7$ | 96.3 | 0.8 | 27.1 | 110.9 | 111.9 | 111 |
| halfcheetah-expert | $\mathbf{128.6} \pm 2.9$ | - | 108.2 | -1.1 | - | - | 107 | 104.8 |
| walker2d-expert | $121.1 \pm 22.4$ | - | 106.1 | 0 | - | - | 125.7 | **153.9** |
| hopper-expert | $\mathbf{135.0} \pm 14.1$ | - | 110.3 | 3.7 | - | - | 109 | 109.9 |

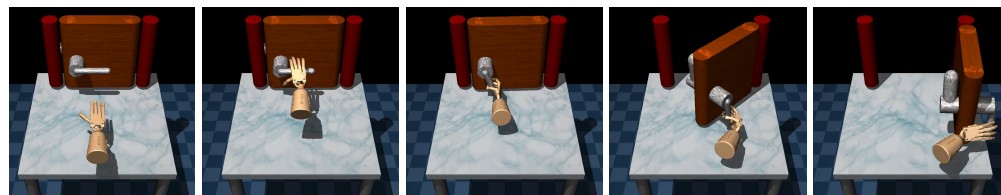

Figure 4: Our learned policies successfully accomplishes manipulation tasks, such as opening a door as shown.

Results are shown in Table 2. UWAC achieves significant improvement on the baseline (BEAR) (Kumar et al., 2019) on all the "human" demonstration datasets, where the datasets lacks state/action diversity and the agent will encounter lots of OOD backups during training. We also obtain state-of-the art performance all other datasets in Adroit.

### 5.4 ANALYSIS OF TRAINING DYNAMICS

Although the baseline method BEAR (Kumar et al., 2019) already improves offline RL training stability on most of the MuJoCo Walkers dataset, we encounter significantly worse training stability when training it on the more complex Adroit hand dataset, especially on demonstrations collected from a narrow policy (i.e. human demonstrations). We show some selected results in Figure 5.

Note that on 5 of the 6 panels shown, the performance of BEAR drops after obtaining peak very early on into training, and sometimes even falls back to initial performance. We also observe similar behavior in all other environments, see full adroit results in Figure 7 in the Appendix. Additionally, we observe strong correlation between the training instability and the explosion of $Q$ values. All performance drops begin at within 5 epochs when $Q$ target estimate greatly exceeds the average return. We attribute the problem of $Q$ function over-estimation and explosion to performing backups from OOD states and actions: As performance improves initially, the OOD $Q$ estimates increases together with the average $Q$ estimates. Since the agent is unable to explore on the OOD actions/states, any over-estimation on the OOD samples can further increase average $Q$ estimates through the Bellman backups, causing a vicious cycle leading to $Q$ value explosion.

In the initial stages of training, the performance of UWAC increases together with the baseline. By down-weighting the OOD backups, UWAC breaks the vicious cycle, and maintains meaningful $Q$ estimates throughout training. This allows UWAC to further train on the offline dataset and surpass BEAR after the performance drop and maintain positive performance.

Table 2: Normalized Average Returns on the D4RL Adroit dataset in the same format as Table 1, over 5 random seeds (± standard deviation). We omit BRAC-p, BRAC-v, cREM, and aDICE because they do not obtain performance meaningful for comparison. We **bold** the highest mean.

| Task Name | UWAC (OURS) | BEAR | BC | SAC-off | CQL($\mathcal{H}$) | CQL($\rho$) | AWR | BCQ | SAC-on |
|---|---|---|---|---|---|---|---|---|---|
| pen-human | $65.0 \pm 3.0$ | -1.0 | 34.4 | 6.3 | 37.5 | 55.8 | 12.3 | **68.9** | 21.6 |
| hammer-human | $\mathbf{8.3} \pm 7.9$ | 0.3 | 1.5 | 0.5 | 4.4 | 2.1 | 1.2 | 0.5 | 0.2 |
| door-human | $\mathbf{10.7} \pm 5.5$ | -0.3 | 0.5 | 3.9 | 9.9 | 9.1 | 0.4 | 0.0 | -0.2 |
| relocate-human | $\mathbf{0.5} \pm 0.6$ | -0.3 | 0.0 | 0.0 | 0.2 | 0.4 | 0.0 | -0.1 | -0.2 |
| pen-cloned | $45.1 \pm 5.8$ | 26.5 | **56.9** | 23.5 | 39.2 | 40.3 | 28.0 | 44.0 | 21.6 |
| hammer-cloned | $1.2 \pm 3.4$ | 0.3 | 0.8 | 0.2 | 2.1 | **5.7** | 0.4 | 0.4 | 0.2 |
| door-cloned | $1.2 \pm 3.6$ | -0.1 | -0.1 | 0.0 | 0.4 | **3.5** | 0.0 | 0.0 | -0.2 |
| relocate-cloned | $\mathbf{0.0} \pm 0.2$ | -0.3 | -0.1 | -0.2 | -0.1 | -0.1 | -0.2 | -0.3 | -0.2 |
| pen-expert | $\mathbf{119.8} \pm 4.1$ | 105.9 | 85.1 | 6.1 | - | - | 111.0 | 114.9 | 21.6 |
| hammer-expert | $\mathbf{128.8} \pm 4.8$ | 127.3 | 125.6 | 25.2 | - | - | 39.0 | 107.2 | 0.2 |
| door-expert | $\mathbf{105.4} \pm 2.1$ | 103.4 | 34.9 | 7.5 | - | - | 102.9 | 99.0 | -0.2 |
| relocate-expert | $\mathbf{108.7} \pm 1.7$ | 98.6 | 101.3 | -0.3 | - | - | 91.5 | 41.6 | -0.2 |

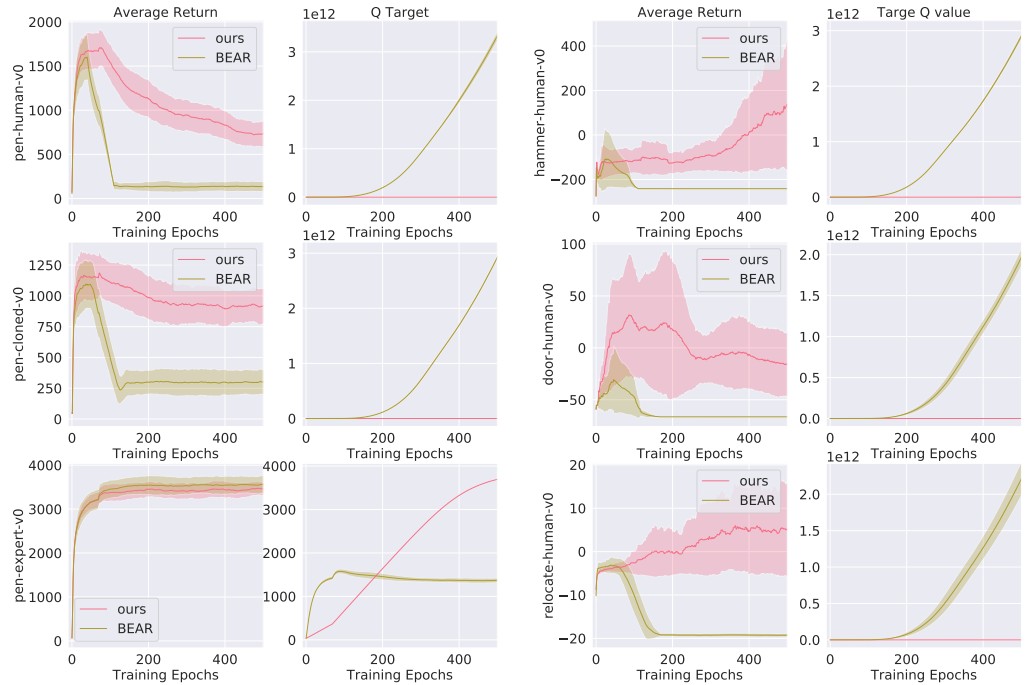

Figure 5: Plot of average return v.s. training epochs, together with the corresponding average Q Target over training epochs. Results are averaged across 5 random seeds. **Left:** Results of different types (human, cloned, expert) on the Adroit pen task. **Right:** Results on human demos on the 3 remaining tasks. The performance of baseline (BEAR) degrades over time (also noted in (Kumar et al., 2019)), and the Target $Q$ value explodes.

## 6 CONCLUSION AND FUTURE WORK

In this work, we have leveraged uncertainty estimation to detect and down-weight OOD backups in the Bellman squared loss for offline RL. We show our proposed technique, UWAC, achieves superior performance and improved training stability, without introducing any additional model or losses. Furthermore, we experimentally demonstrate the effectiveness of dropout uncertainty estimation at detecting OOD samples in offline RL. UWAC also can be applied to stabilize other actor-critic methods, and we leave the investigation to future works.

In addition, our work demonstrates a valuable application of Bayesian uncertainty estimation in RL. Future works can combine model-based and model-free methods for offline or off-policy RL and use uncertainty estimation to decide when to use the model to train the actor. Additionally, uncertainty estimation may be used to guide curiosity based RL agents for on-policy curiosity-based learning.

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

# A APPENDIX

## A.1 ANALYSIS FOR CONVERGENCE PROPERTIES

We firstly state **Theorem B.1** in (Kumar et al., 2019) as Theorem A.1 for the sake of completeness.

**Theorem A.1.** *Suppose we run approximate distribution-constrained value iteration with a set constrained backup $\mathcal{T}^\Pi$ on a set of policies $\Pi$. Let $\delta(s,a)$ be the upper-bound for the Bellman approximation error for a given state-action pair $(s,a)$ over $k$ training steps: $\delta(s,a) = \sup_k \left| Q_k(s,a) - \mathcal{T}^\Pi Q_{k-1}(s,a) \right|$. Then,*

$$\lim_{k \to \infty} \mathbb{E}_{\rho_0} \left[ |V_k(s) - V^*(s)| \right] \leq \frac{\gamma}{(1-\gamma)^2} \left[ C(\Pi)\mathbb{E}_\mu \left[ \max_{\pi \in \Pi} \mathbb{E}_\pi [\delta(s,a)] \right] + \frac{1-\gamma}{\gamma}\alpha(\Pi) \right]$$

*with the suboptimality constant ($\alpha(\Pi)$) and the concentrability coefficient defined as:*

$$\alpha(\Pi) = \max_{s.a} \left| \mathcal{T}^\Pi Q^*(s,a) - \mathcal{T}Q^*(s,a) \right| \; ; C(\Pi) \stackrel{def}{=} (1-\gamma)^2 \sum_{k=1}^{\infty} k\gamma^{k-1} c(k)$$

The proof of the theorem is a direct modification of the contraction proof in Theorem 3 of (Farahmand et al., 2010) or Theorem 1 of (Munos, 2003).

The *suboptimality constant* ($\alpha(\Pi)$) captures how far $\pi^*$ is from $\Pi$, namely the suboptimality of the actor. The *concentrability coefficient* quantifies how far the visitation distribution generated by policies from $\Pi$ is from the training data distribution, namely the degree to which the training may encounter OOD actions and states. Kumar et al. (2019) note a trade-off between $\alpha(\Pi)$ and $C(\Pi)$, and propose to bound both terms by constraining $\Pi$ to the set of policies with support the same as the training set policy with MMD loss.

However, the most important Bellman approximation error term which is the root cause of the bootstrapping problem is still left unbounded. We proceed to show that for $\pi'(a|s) = \frac{\beta}{\sup_k \sqrt{Var[Q_k(s,a)]}}\pi(a|s)/Z$. Assuming that $Z \geq 1$, and that $Q$ is bounded, we can bound the Bellman error term $\max_{\pi'} \mathbb{E}_{\pi'}[\delta(s,a)]$ by any constant $C$ with arbitrarily high probability by optimizing on $\pi'$.

Note that Theorem A.2 considers down-weighting by inverse the square-root of the variance (standard deviation), which is different from the inverse variance in Equation 3,4,5 and Algorithm 1. Down-weighing by the variance has the same practical effect since we clip the ratio for numerical stability. We adopt variance for practical implementation for the ease of tracing after multiple max,min,summation operations in Algorithm 1.

**Theorem A.2.** *Let $\pi'(a|s) = \frac{\beta}{\sup_k \sqrt{Var[Q_k(s,a)]}}\pi(a|s)/Z(s)$, with the normalization factor $Z(s) = \int_a \frac{\beta}{Var[Q(s,a)]}\pi(a|s)$ as in equation 3. Assume that 1) $\forall s : Z(s) \geq 1$ and 2) $Q$ is bounded ($\forall s,a : |Q(s,a)| \leq Q_m$).*

*Then for any $C \in \mathbb{R}$, there exists $\beta, K$ such that*

$$P\left( \max_{\pi'} \mathbb{E}_{\pi'}[\delta(s,a)] \geq C \right) \leq \frac{1}{K^2}$$

*Proof.* We firstly apply triangle inequality to unwrap the original formulation into a sum of two differences, and bound the two terms respectively.

$$\max_{\pi'} \mathbb{E}_{\pi'}[\delta(s,a)] = \max_{\pi'} \mathbb{E}_{\pi'} \left[ \sup_k \left| Q_k(s,a) - \mathcal{T}^\Pi Q_{k-1}(s,a) \right| \right]$$

$$= \max_{\pi'} \mathbb{E}_{\pi'} \left[ \sup_k \left| Q_k(s,a) + E[Q_k(s,a)] - E[Q_k(s,a)] - \mathcal{T}^\Pi Q_{k-1}(s,a) \right| \right]$$

$$\leq \max_{\pi'} \mathbb{E}_{\pi'} \left[ \sup_k |Q_k(s,a) - E[Q_k(s,a)]| \right] + \max_{\pi'} \mathbb{E}_{\pi'} \left[ \sup_k \left| E[Q_k(s,a)] - \mathcal{T}^\Pi Q_{k-1}(s,a) \right| \right]$$

Starting with the red term, we firstly obtain a probabilistic bound on the distance term inside the expectation with the Chebyshev's inequality

$$P\left(|X - E[X]| \geq \sigma K\right) \leq \frac{1}{K^2}$$

$$P\left(|Q_k(s,a) - E[Q_k(s,a)]| \geq K\sqrt{Var[Q_k(s,a)]}\right) \leq \frac{1}{K^2}$$

$$P\left(\frac{\beta}{\sup_k \sqrt{Var[Q_k(s,a)]}} |Q_k(s,a) - E[Q_k(s,a)]| \geq \beta K\right) \leq \frac{1}{K^2}$$

Secondly, note that by assumption $|Q|$ is bounded by $Q_m$. This provides us an upper-bound on the difference $|Q(s,a) - E[Q(s,a)]| \leq 2Q_m$. Making use of both the general upper-bound and the tight probabilistic bound, by setting $\pi'(a|s) = \frac{\beta}{\sup_k \sqrt{Var[Q_k(s,a)]}}\pi(a|s)/Z(s)$, we have

$$\max_{\pi'} \mathbb{E}_{\pi'}\left[\sup_k |Q_k(s,a) - E[Q_k(s,a)]|\right] = \max_{\pi'} \mathbb{E}_{\pi}\left[\frac{\beta}{\sup_k \sqrt{Var[Q_k(s,a)]}} \sup_k |Q_k(s,a) - E[Q_k(s,a)]|/Z(s)\right]$$

$$\leq \left(1 - \frac{1}{K^2}\right)\beta K + \frac{2}{K^2}Q_m \leq B$$

By assumption $Z(s) \geq 1$ and can be safely ignored. By picking the appropriate $K$ and $\beta$, we can bound the red term by any constant $B \in \mathbb{R}$. The same bound holds for the blue term since $E[\mathcal{T}^\Pi Q_{k-1}(s,a)] = E[Q_k(s,a)]$. We therefore arrive at a constant bound for the Bellman error term $\max_{\pi'} \mathbb{E}_{\pi'}[\delta(s,a)]$. $\qquad\square$

Note that in Theorem A.2 Assumption 1) does not change the optimization problem in equation 4, 5 and Assumption 2) can be easily satisfied by imposing Spectral Norm on the $Q$ function.

Now according to the constant bound on $\delta(s,a)$ from Theorem A.2, the convergence of our proposed framework follows directly from Theorem A.1 (Kumar et al., 2019; Farahmand et al., 2010; Munos, 2003), with the set of policies $\Pi' = \left\{\pi' \mid \pi'(a|s) = \frac{\beta}{\sup_k \sqrt{Var[Q_k(s,a)]}}\pi(a|s)/Z(s), \pi \in \Pi\right\}$.

## A.2 IMPLEMENTATION DETAILS

**LunarLander:** We set our expert to be a simple 3-layer actor-critic agent trained to completion with (Peng et al., 2019). We take the final replay buffer (size 100,000) with average reward of 269.7. The vertically clipped dataset in Figure 6 contains 76,112 samples, and the horizontally clipped dataset in Figure 3 contains 21,038 samples.

We then train a simple 3-layer actor-critic off-policy agent on the clipped datasets according to Algorithm 1 (we do not take the MMD loss in line 11 to enlarge the effect of OOD samples).

**Baseline (BEAR):** We ran benchmarks on the official GitHub code[2] of BEAR and the updated version[3] provided by the authors. We ran parameter search on all the recommended parameters kernel_type∈{gaussian, laplacian}, mmd_sigma∈{10,20}, 100 actions sampled for evaluation, and 0.07 being the mmd_target_threshold. We are able to reproduce the results reported in (Fu et al., 2020) with both the official GitHub and the updated version.

**Our method (UWAC):** We apply our weighted loss to Algorithm 1 to the updated BEAR code provided by Kumar et al. (2019). We keep the hyper-parameters and the network architecture exactly the same as in BEAR. For experiments on the Adroit hand environment, we further enforce Spectral Norm on the Q function for better stability similar to (Yu et al., 2020) and theoretical guarantee as

---

[2]github.com/aviralkumar2907/BEAR
[3]github.com/rail-berkeley/d4rl_evaluations

shown in Appendix A.1. We clip the inverse variance to within the range of $(0.0, 1.5)$ for numerical stability. For the choice of $\beta$ in Algorithm 1. We swept over 3 beta values from the set $\{0.8, 1.6, 2.5\}$, determined by matching the average uncertainty output during training time. We found that the model is quite robust against betas: $0.8, 1.6$ gave similarly good performance across all tasks in our experiments. We also note that $\beta$ can be absorbed into the learning rate since it acts both on the actor loss and critic loss. However, since the MMD loss from BEAR is not $\beta$-weighted, we make the design choice to tune $\beta$ in stead of the MMD weight $\alpha$.

**Ablations:**

Our first study isolates the effect of Spectral Norm on agent performance. Although BEAR + Spectral Norm enforces a bounded $Q$ function and maintains good training stability, Spectral Norm does not handle OOD backups on the narrow Adroit datasets. We discover experimentally that BEAR+SN performs much worse than BEAR only, we plot the complete results of BEAR+SN v.s. BEAR in Figure 9.

Our second study isolates the effect of Dropout on agent performance as a regularizer, since dropout alone does not handle OOD backups on the narrow Adroit datasets. We observe experimentally that UWAC without uncertainty weighing (BEAR+Dropout+Spectral Norm) does not change the behavior of BEAR under Spectral Norm (Figure 10) and performs worse than UWAC (Figure 11) and the original BEAR (Figure 12).

## B FIGURES

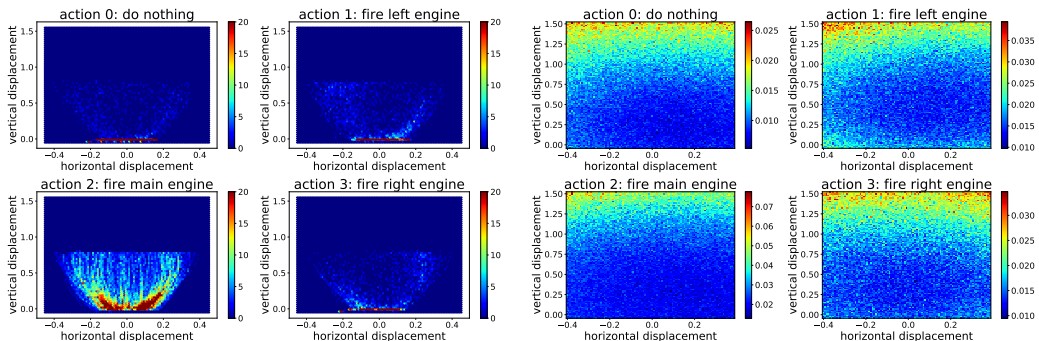

Figure 6: **Left.** The training dataset has observations with vertical displacements $> 0.8$ removed. This makes all states on the top OOD states. **Right.** Our model estimates higher uncertainty (brighter color) on the top and lower uncertainty (colder color) on the bottom.

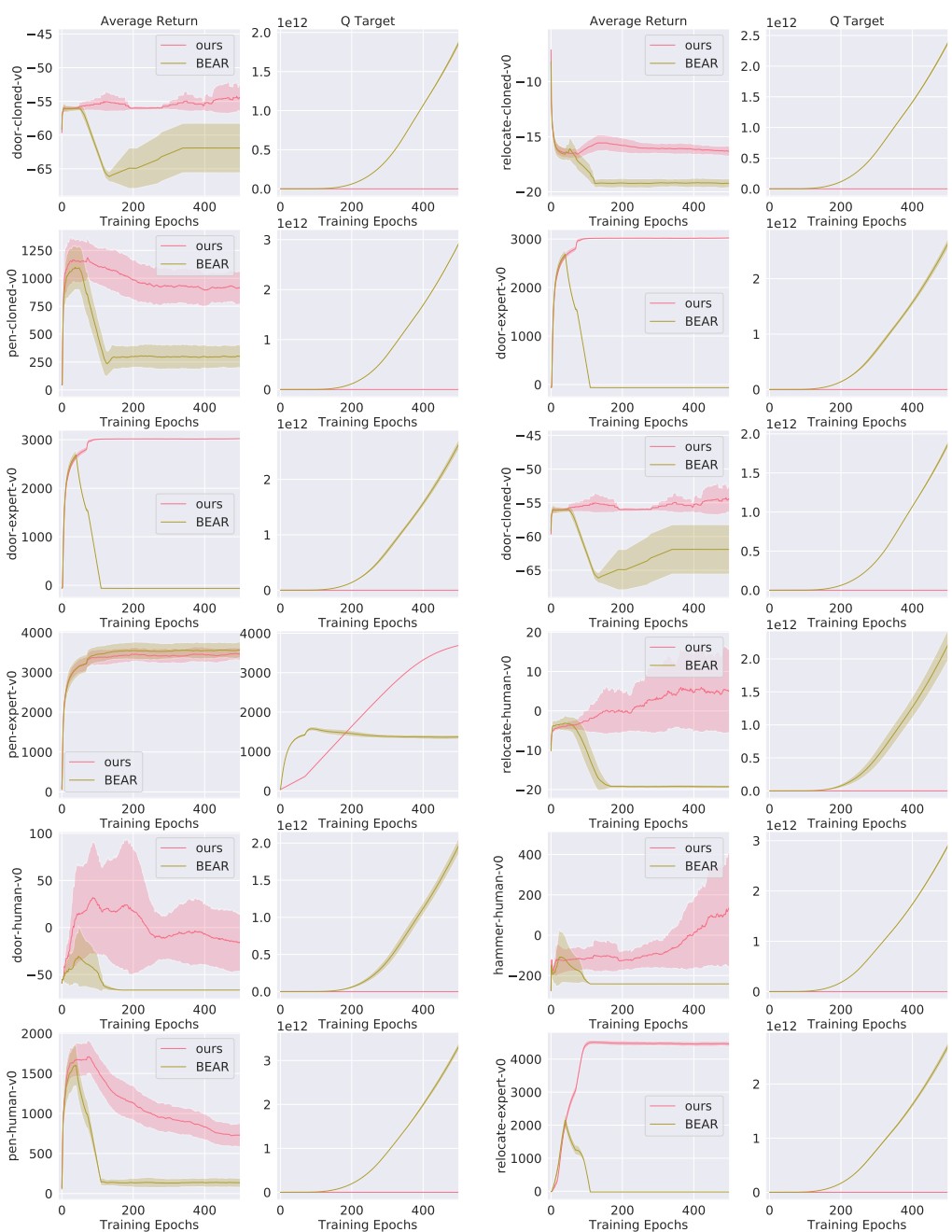

Figure 7: Plot of average return v.s. training epochs, together with the corresponding average Q Target over training epochs on the D4RL Adroit hand offline data set. Results are averaged across 5 random seeds. Note that the performance of baseline (BEAR) degrades over time (also noted in original paper Kumar et al. (2019)), and the Target $Q$ value explodes. Our method, UWAC, achieves significantly better overall performance and training stability.

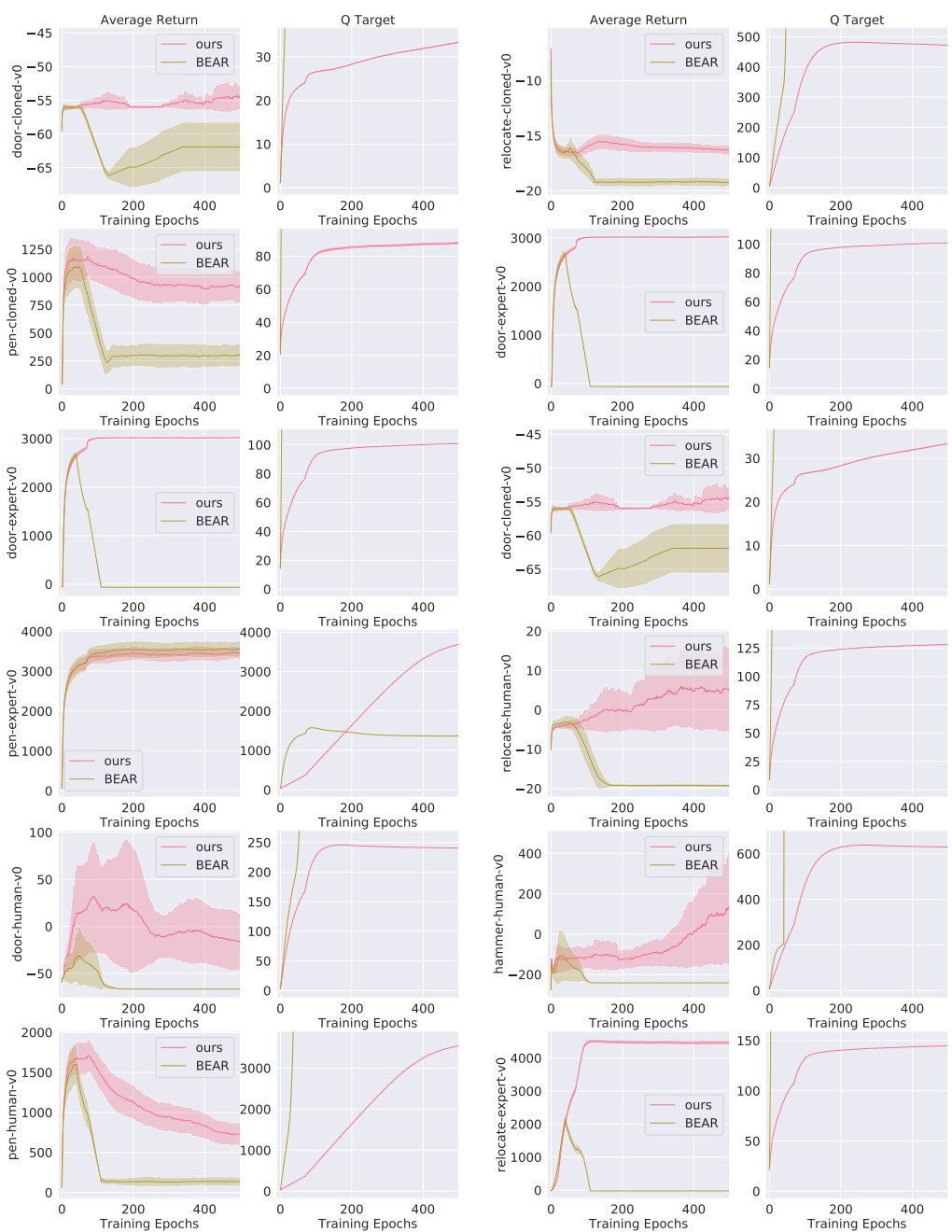

Figure 8: Plot of average return v.s. training epochs (zoomed-in). The figure is the same as 7, except that the second column is zoomed-in on the Q values of the UWAC critic.

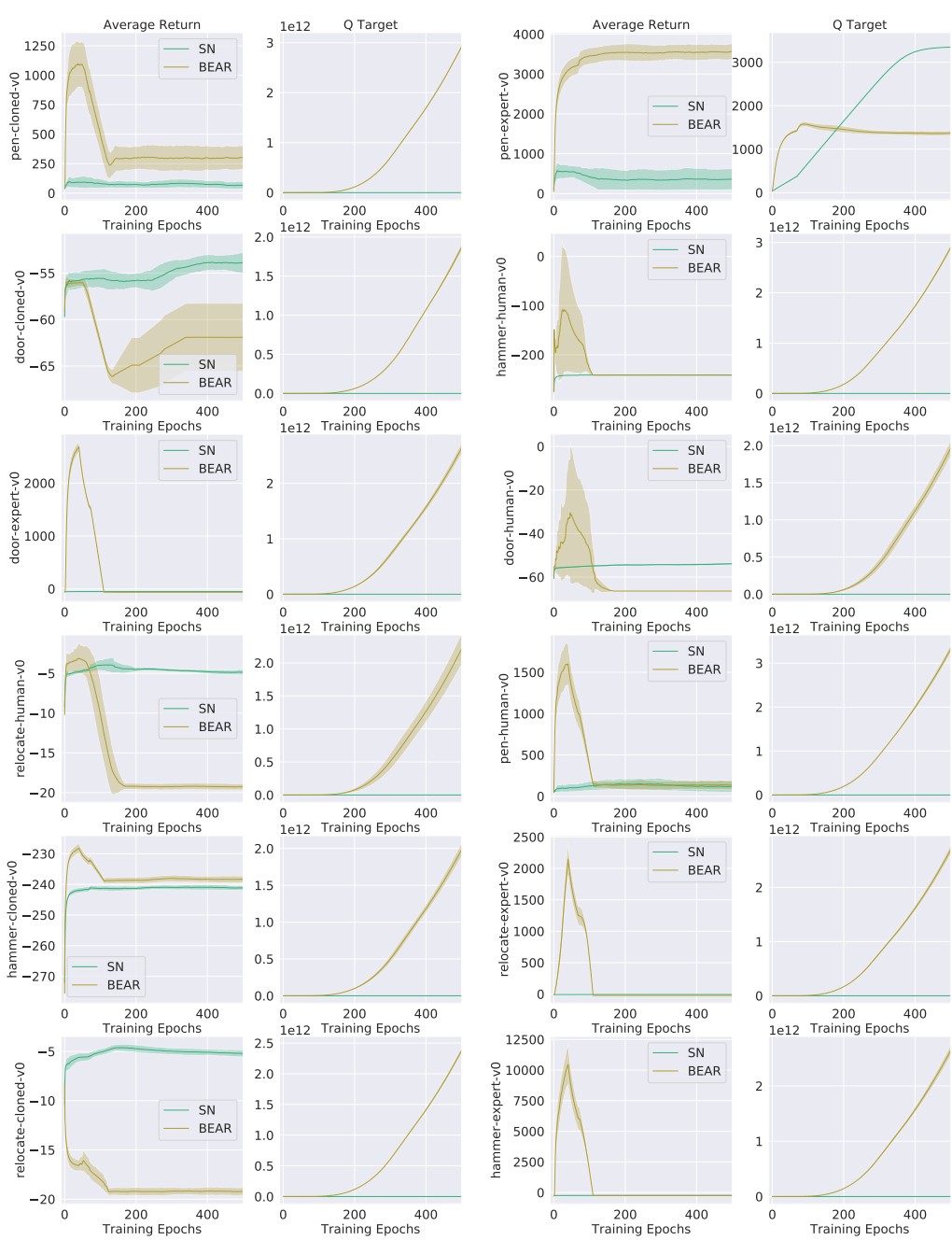

Figure 9: **Ablation:** Plot of average return v.s. training epochs for BEAR v.s. BEAR+Spectral Norm, together with the corresponding average Q Target over training epochs on the D4RL Adroit hand offline data set. Results are averaged across 5 random seeds. Although BEAR with Spectral Normalized $Q$ function maintains stable $Q$ estimate during training, BEAR with SN achieves significantly worse training performance in terms of average return.

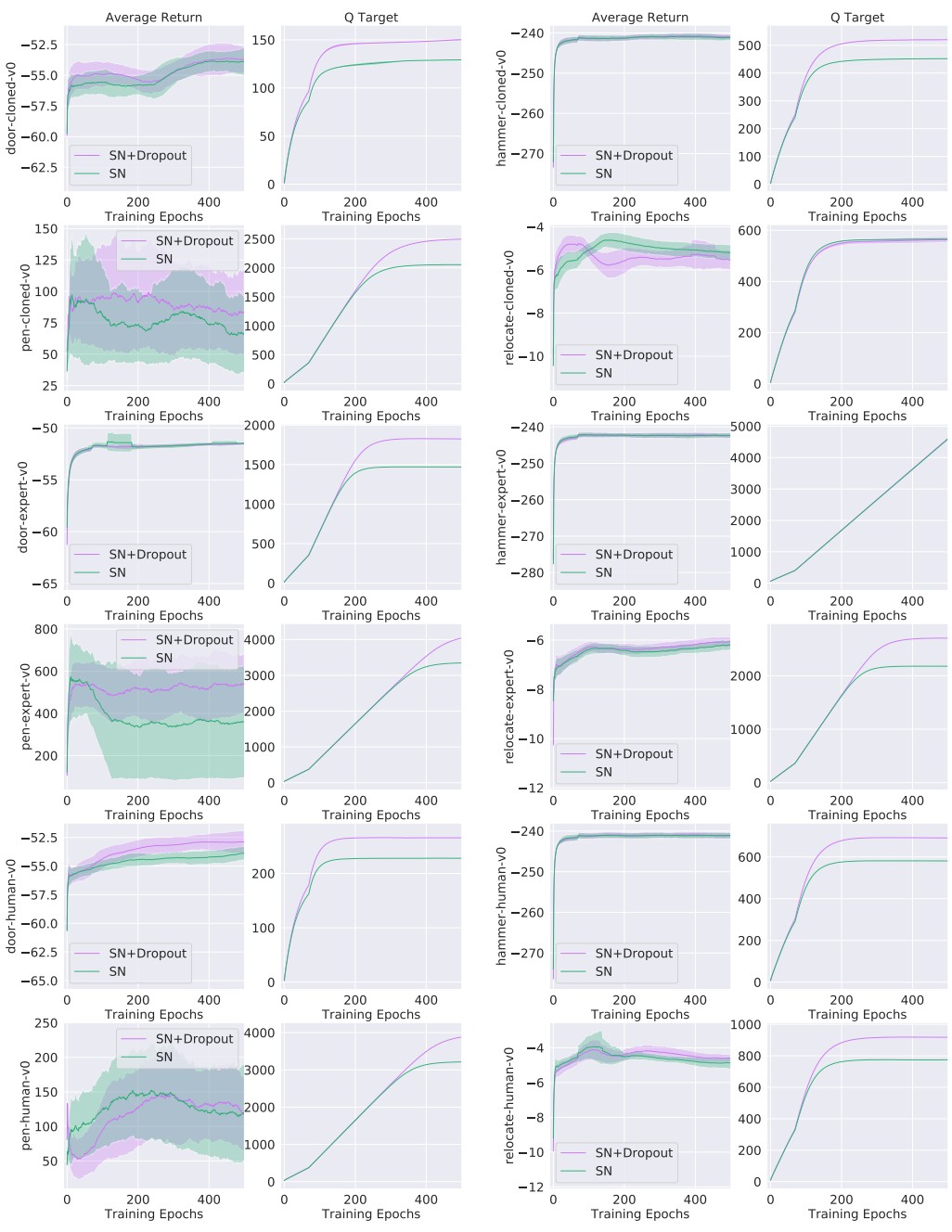

Figure 10: **Ablation:** Plot of average return v.s. training epochs for BEAR+Spectral Norm v.s. BEAR+Dropout+Spectral Norm, together with the corresponding average Q Target over training epochs on the D4RL Adroit hand offline data set. The results are averaged across 5 random seeds. Without the UWAC reweighing loss, performing dropout on the Q function does not lead to improved performance.

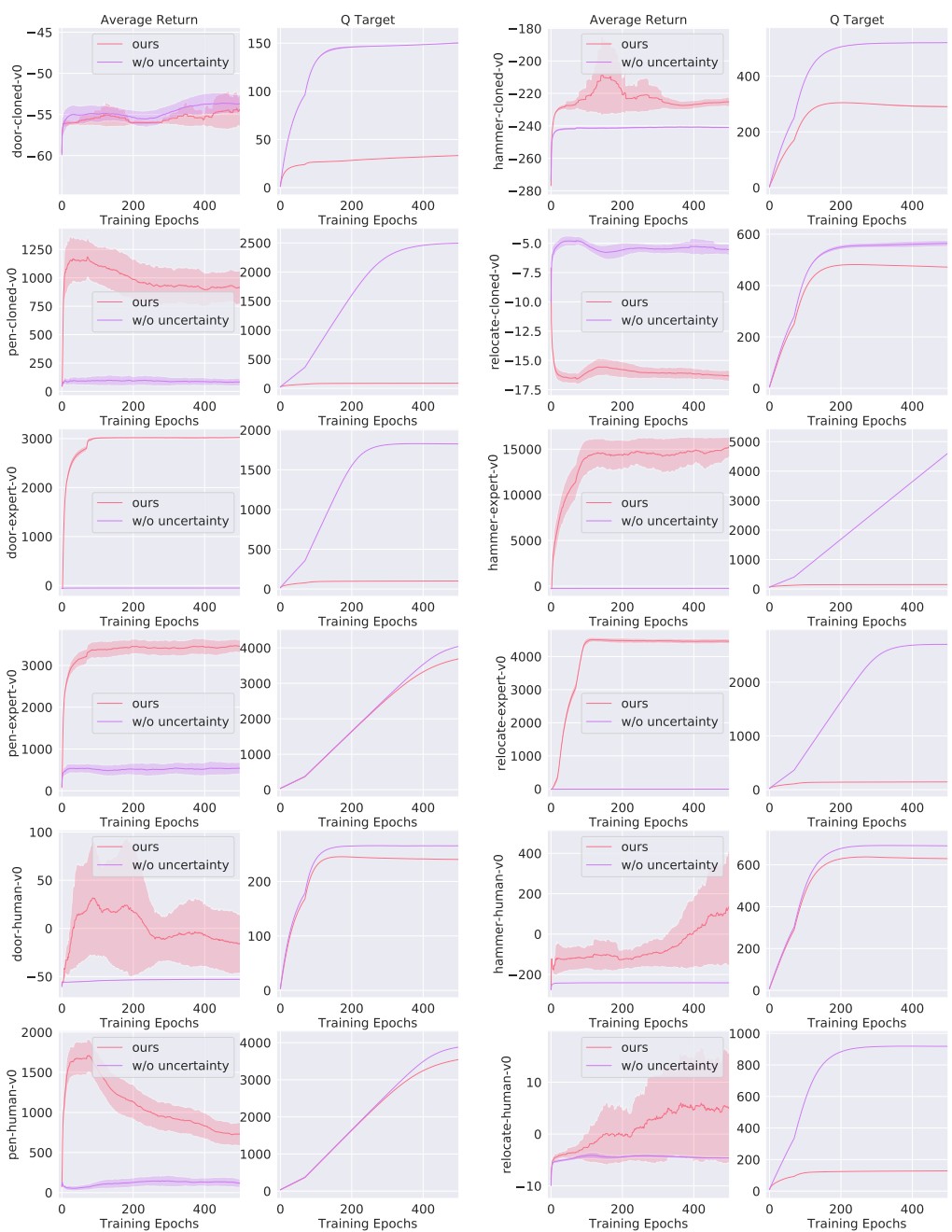

Figure 11: **Ablation:** Plot of average return v.s. training epochs for UWAC (ours) v.s. ours without uncertainty weighing but with dropout in the Q function, together with the corresponding average Q Target over training epochs on the D4RL Adroit hand offline data set. The results are averaged across 5 random seeds. Without the weighing loss, performance of the agent drops drastically. Note that low performance on hammer-cloned, door-cloned, and relocated-cloned may be attributed to the bad quality of the datasets caused by data collection (explained in section 5.3)

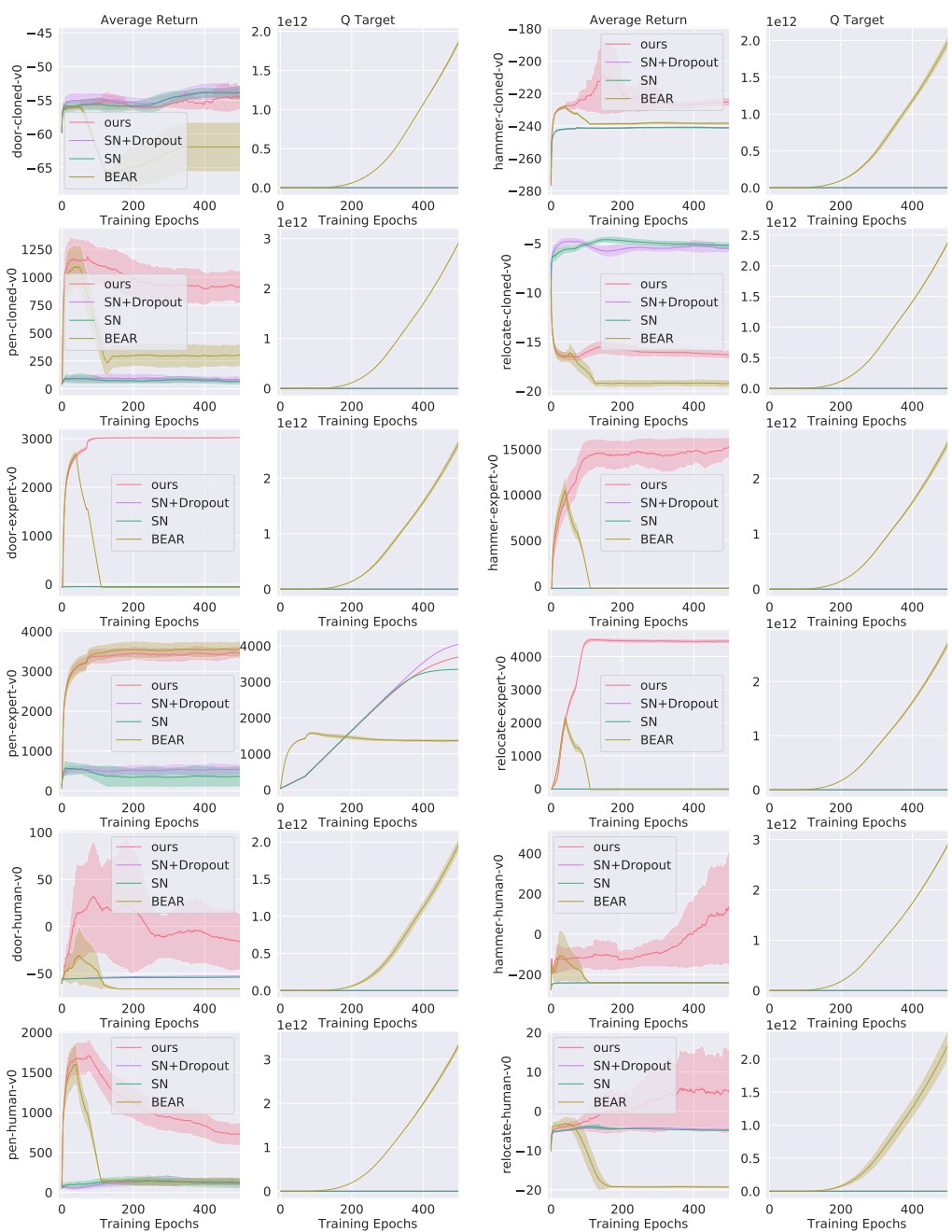

Figure 12: **Ablation:** Figure 9, 10, 11 plotted together. Note that SN+Dropout (purple) is also denoted as ours-w/o-uncertainty in Figure 11.

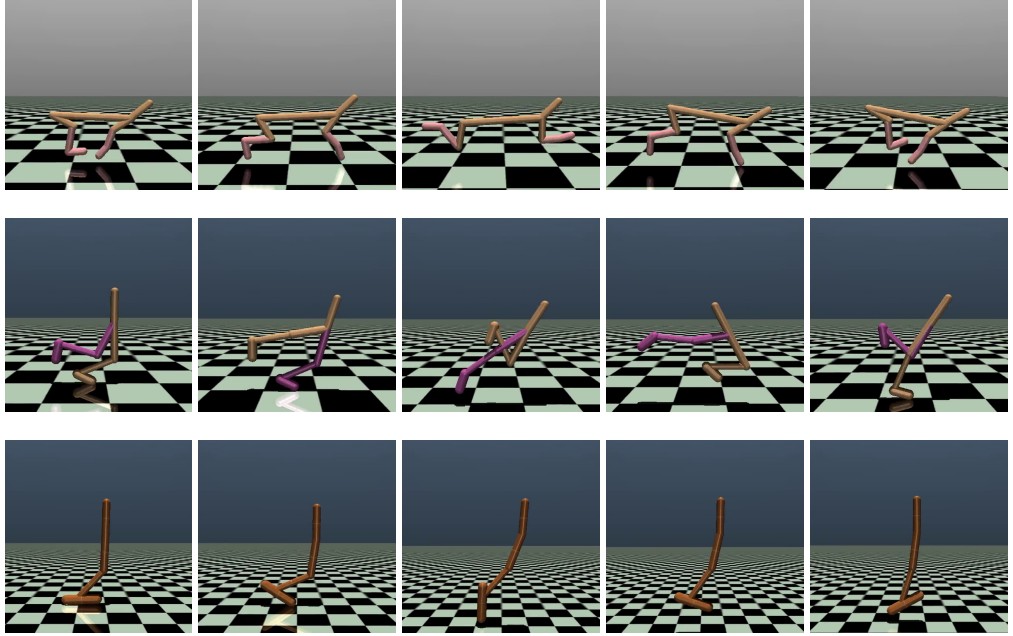

Figure 13: Sequences of our offline agent trained from expert demonstrations executing learned policies performing on the halfcheetah, walker2d, and hopper tasks in the MuJuCo Gym environment. See the videos attached in the supplementary.

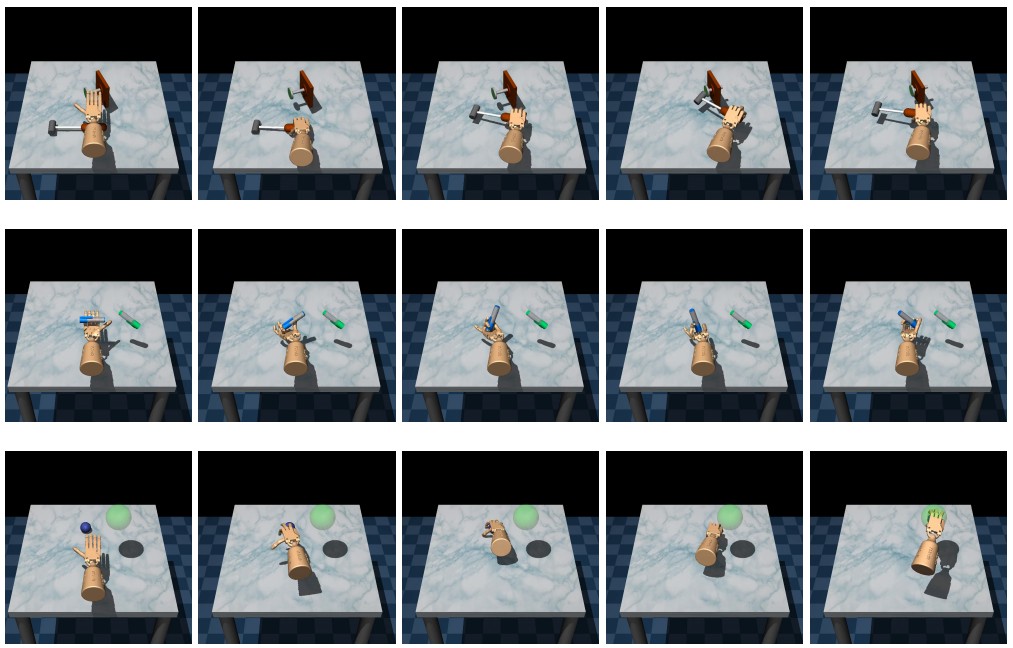

Figure 14: Sequences of the agent trained from human demonstrations executing learned policies performing the Adroit tasks of hammering a nail, twirling a pen and picking/moving a ball. The task of opening a door is shown in Figure 4. See the videos attached in the supplementary.

