# OpenReview forum: "Uncertainty Weighted Offline Reinforcement Learning"
_ICLR.cc/2021/Conference — Reject_

### Official Review · AnonReviewer3 · 2020-10-19
**Simple, empirically working solution for offline RL**

**Rating:** 5
**Confidence:** 4

**Review:**

This paper proposes a simple offline RL algorithm, called UWAC, which down-weights the bellman errors for uncertain state and action pair while the uncertainty is measured by MC-dropout. The algorithm empirically shows competitive performance in various test domains in the D4RL benchmarks, including Adroit hand manipulation tasks. Overall, I agree with the main motivation for the paper that tries to estimate and use uncertainty, and it seems that the method empirically achieves the goal. Still, I have a concern about the theoretic validity of MC-dropout in the Q-learning setup and the clarity of the papers (especially typos and proofs in Appendix). Hopefully, the authors can address my concern in the rebuttal period.

Concerns:
* It is unclear whether MC-dropout measures epistemic uncertainty precisely even in Q-learning in which bootstrapping is in use; since the target Y for certain input X depends on its own estimation, MC-dropout could become unreliable. In the meantime, the same algorithm could leverage uncertainty measured for transition dynamics with proper scaling as it is done for $Q$. Why can measuring epistemic uncertainty for $Q$ via MC-dropout be valid (or effective)? What is the main advantage (or motivation) over other uncertainty measuring methods (such as deep ensemble) or other domains (such as transition dynamics learning)?
* Is there a fundamental reason for using BEAR as a base algorithm to work upon? How well will the raw UWAC -- without MMD penalty of BEAR -- perform? As it is mentioned in Section 3.2, the support constraints via MMD is rather empirical and could be wrong in some cases. If raw UWAC fails, then what will be the main cause?

Minor Comments:

* Section 4.1, Variance equation: what is $\sigma^2$, what is the theoretic impact of this term, and how is this value decided from the implementation perspective? Also, why $\hat{Q}$ is in a vector notation (transpose)?
* Algorithm 1: is $Z$ approximated with $\{a_i\}$, or is $Z$ simply ignored?
* How to set a proper $\beta$, and how sensitive is UWAC to the hyperparameter?
* Section 5.1: what do you mean by the ‘final’ replay buffer? Does this mean that the last 100,000 transition tuples are saved for the offline RL dataset? Then, why do trajectories gathered by executing nearly expert policy have relatively complete coverage over the observation space? The expert policy won’t execute the action that is suboptimal, so the coverage would be limited. Also, it would be nice if you can include the uncertainty figure of Q which is trained without skewing.
* Figure 5, pen-human-v0 results: what is the possible reason for the degeneration of UWAC? Wrong OOD detection?

Typos:

* Equations are the components of a sentence, and every sentence ends with proper punctuation; add proper punctuation before or after every equation.
* Section 3.1 -- $\gamma \in (0,1)$ → $\gamma \in (0,1]$.
* The second paragraph of page 5 -- erase parenthesis around $\pi$.
* For the completeness of the paper, please include all loss functions used in the algorithm; the exact MMD penalty term is missing in Algorithm 1.
* Target network update should be $\theta’ \leftarrow (1-\tau) \theta’ + \tau \theta$, if $\tau$ is a polyak coefficient used for slowly moving target network. Also, just use $\theta_{1,2}$ instead of $i$. $i$ is used in another context, so this is confusing.
* In Algorithm 1, $\phi_i$ in line 12 is a typo, and $\{ a_j \sim D\}_{i=1}^{n=10}$ should be $\{a_j \sim D\}_j^n$.
* In Tables 1 and 2, when your results are the best, the results that lie in some confidence interval of yours should also be bolded for a fair comparison.
* The second paragraph from the bottom of page 7, “when Q Target” → “when Q target”, “and actions: As performance” → “and actions: as performance”.
* It would be nice if Figures 7 and 8 in the appendix can be integrated into a single figure.

---

> ### Author Response · Authors · 2020-11-13
> **Addressing the concerns on 1) Epistemic v.s. Aleatoric uncertainty.   2) MC-dropout on Q.   3) BEAR as baseline.     and     Clarifications for minor comments**
>
> Thank you for the detailed comments, especially on helping to improve the presentation of the paper. We have fixed the issues you pointed out and below are our responses to your concerns:
>
> [MC-dropout on Q function for uncertainty estimation] MC dropout on Q function has already been studied and applied to online RL to encourage exploration through Thompson Sampling (Section 5.4 of [2]). Given that the method is so simple, it certainly has some degrees of limitations. However, random prior based methods similar to dropout have been widely applied to capture uncertainty in RL [2,3,4,5,6,7]. In addition, our experiments section 5.1 empirically demonstrates that dropout uncertainty estimation can identify OOD states, which correspond to epistemic uncertainty estimation.
>
> [Epistemic v.s. Aleatoric] Aleatoric uncertainty captures noise inherent in the observations. On the other hand, epistemic uncertainty accounts for uncertainty in the model – uncertainty which can be explained away given enough data [1]. Dropout as Approximate Bayesian Inference is widely used to model the epistemic uncertainty [1,2]. Dropout uncertainty estimation has also been applied in online RL to encourage exploration in unknown states through Thompson Sampling (Section 5.4 of [2]). Our experiments section 5.1 also empirically demonstrates that dropout uncertainty estimation can identify OOD states, which correspond to epistemic uncertainty estimation.
>
> [BEAR as baseline] UWAC and BEAR MMD loss acts complementary to each other. We would like to note two points: 1) The support constraint on the actor is necessary. Reweighting proposed by UWAC does not change the support of the actor, therefore some hard OOD state-action pairs are avoided with a support-based loss like MMD loss. 2) The support constraint on the actor can be relaxed. A too strong MMD penalty can be harmful, but UWAC can allow us to set a relatively low weight by setting a low \alpha in Algorithm 1.
>
> [1] Kendall, Alex, and Yarin Gal. "What uncertainties do we need in bayesian deep learning for computer vision?." NeurIPS. 2017.
>
> [2] Gal, Yarin, and Zoubin Ghahramani. "Dropout as a bayesian approximation: Representing model uncertainty in deep learning." ICML. 2016.
>
> [3] Osband, Ian, John Aslanides, and Albin Cassirer. "Randomized prior functions for deep reinforcement learning." Advances in Neural Information Processing Systems. 2018.
>
> [4] Fortunato, Meire, et al. "Noisy Networks For Exploration." International Conference on Learning Representations. 2018.
>
> [5] Lipton, Zachary C., et al. "Bbq-networks: Efficient exploration in deep reinforcement learning for task-oriented dialogue systems." arXiv preprint arXiv:1608.05081 (2016).
>
> [6] Tang, Yunhao, and Alp Kucukelbir. "Variational deep q network." arXiv preprint arXiv:1711.11225 (2017).
>
> [7] Touati, Ahmed, et al. "Randomized value functions via multiplicative normalizing flows." Uncertainty in Artificial Intelligence. PMLR, 2020.
>
> Clarifications:
> Minor Comments:
>
> 1. \sigma^2 is the observation noise, or the aleatoric uncertainty, with detailed explanations in [1]. This can be captured by learning Q function as a distribution and performing MAP inference.
> 2. Z is absorbed into hyper-parameter \beta
> 3. The ‘final’ replay buffer in our case is indeed an expert replay buffer. However, the states have relatively complete coverage over observation space because the lander initiate with a different movement, causing the lunar lander to have a different movement direction/speed, and therefore varying the landing pattern. The variations are built into the default gym environment.
> 4. We believe the degeneration comes from the intrinsic difficulties learning from the human demonstrations alone. Each “human” dataset consists of only 15 trajectories collected from people performing the task in VR. The demonstrations may not even have enough coverage over the state space. The intrinsic difficulties and inability to explore makes the learning difficult and unstable.

---

### Official Review · AnonReviewer2 · 2020-10-27
**Official Blind Review**

**Rating:** 8
**Confidence:** 4

**Review:**

The authors present a simple and efficient method for training offline RL agents. They estimate the epistemic uncertainty of their model through dropout  variational inference. While this method is not novel, it has never been applied to offline RL as far as I know. On the basis of this epistemic uncertainty estimate, they regularize the policy search to avoid sensitivity to overestimates in poorly known states. The way this regularization is performed is taken from [Kumar2019]. The authors then experimentally demonstrate the effectiveness of dropout uncertainty estimation for RL, and achieve the best performance on a classic offline RL benchmark, with the best offline RL algorithms for continuous state-action MDPs.

There is not much to say about the paper. It is well written and positioned. One could that the method is the straigthforward application of dropout variational inference to [Kumar2019] algorithm, but it remains that the empirical results are quite significantly improving the state of the art, and as such, it deserves to be known.

I only have one minor remark to share: Section 4.1, the explanation of the variance decomposition is a bit misleading. Red minus blue is the dropout variational variance, and this is what measures the uncertainty of the model (not singlehandedly the red term).

---

> ### Author Response · Authors · 2020-11-13
> **Follow-up on suggestions for clarification**
>
> Thank you for the encouraging comments. We are encouraged by the appreciation that 1) our method is well-motivated, 2) our straightforward algorithm achieves good performance, and 3) the paper is well written.
>
> We have updated Section 4.1 to improve the explanation.

---

### Official Review · AnonReviewer1 · 2020-10-28
**A simple dropout method for offline RL with uncertainty estimation**

**Rating:** 7
**Confidence:** 4

**Review:**

##########################################################################

Summary:


This paper shows that a easy-to-implement dropout to estimate uncertainty can address the value overestimation problem in offline RL for OOD action or states. The paper baselined against the results from D4RL and showed to outperform those more than half of the times.


##########################################################################
Pros:

Uncertainty is an important yet underexplored domain in Offline RL. Having a simple method that can be applied to actor-critic based methods is attractive.

As far as I know, there hasn’t been literature that combines dropout and Offline RL uncertainty estimation in the loss function

The performance of the algorithm is decent on D4RL, a standard offline RL baseline.

##########################################################################

Cons:

Since offline RL is motivated by a potentially costly evaluation and/or data collection process, It is risky to add additional hyperparameters in offline RL. Thus it would be informative to mention the following in the paper: Is your model sensitive to the choice of beta? How many values of beta did you sweep over?


##########################################################################

Questions during rebuttal period:
1. I could be missing something. Since the beta hyperparameter is a constant in all losses. Can it be absorbed into the learning rate?

2. In the appendix, I found theorem A.2 confusing. It assumes that Q is bounded for any (s,a). this assumption makes the proof almost trivial. The given justification is that Q can be bounded with spectral norm, yet in practice spectral norm performs much worse than no spectral norm. I thus find the theoretical backing shaky, especially you further assumed in the paper that Z can be absorbed into beta. Can you expand more on this?

3. Could you address the Cons above?

##########################################################################

Additional suggestions:

1. Can you add a graph in the appendix where you zoom in a bit on the Q target? I want to see whether it still overestimates, and if so by how much. Having the y axis = 1e12 hides those information.

---

> ### Author Response · Authors · 2020-11-13
> **Addressing the concerns on 1) Hyper-parameter beta.  2) Theorem A.2.  3) Absorb Z into beta.  4) Zoom in on Q-target**
>
> Thank you for the positive comments that our proposed method is 1) simple, 2) well-motivated from uncertainty estimation, and 3) achieves good performance on a variety of tasks.
>
> [Hyper-parameter beta] First of all, as we show in the experiments, introducing beta is essential for training stability. We swept over 3 beta values from the set {0.8, 1.6, 2.5}, determined by matching the average uncertainty output during training time. We found that the model is quite robust against betas: 0.8, 1.6 gave similarly good performance across all tasks in our experiments. Therefore, the process of searching for a beta should not be challenging. We have updated the manuscript to reflect the choice of beta.
> Also, the decision to include beta is a design choice. As you have pointed out, beta can technically be absorbed into the learning rate since it acts both on the actor loss and critic loss. However, since the MMD loss from BEAR is not beta-weighted, we will need to tune the MMD weight alpha to reflect the change. Thank you for making this observation, we have updated the manuscript to reflect this.
>
> [Theorem A.2] The assumption that Q is bounded does not make proof trivial. We only assume that there EXISTS an upper-bound, the upper-bound can be arbitrarily large and can differ from environment to environment. And since there are only limited resources in the real world, the reward needs to be finite in most cases with the discount factor (<1), and therefore one can assume that the Q function is bounded. Additionally, boundedness can be easily enforced via spectral normalization. We observe that although BEAR with spectral normalization only performs worse than BEAR in terms of peak performance, most trials have stopping performance (at iteration 500) the same as or higher than BEAR (Figure 8).
>
> [Absorb Z into beta] The decision to absorb Z(s) into beta is indeed an experimental relaxation, since true integral is intractable. We performed experiments on D4RL MuJoCo Walkers dataset with monte-carlo estimates for the integral Z(s) by sampling actions from the actor, and observed no difference in performance.
>
> [Zoom in on Q-target] See Figure 8 in the updated appendix.

---

> > ### Comment · AnonReviewer1 · 2020-11-23
> > **Clarifying the comment around Theorem A.2**
> >
> > [Hyper-parameter beta] Thank you for expanding on the model’s robustness against beta values and for pointing out the MMD loss is not beta-weighted. That makes sense.
> >
> > [Theorem A.2] The reason I said “The assumption that Q is bounded makes the proof trivial.” is because, as you rightfully pointed out in A.1, “the most important Bellman approximation error term which is the root cause of the bootstrapping problem is still left unbounded.” However bounding Q alone is a sufficient condition for the bellman approximation error becoming bounded, even without reweighing the policy by the standard deviation or the variance. If that’s true and both reweighed and not-reweighted policy converges according to Theorem A.1, then the key to better performance is not whether they converge or not, but perhaps the tightness of the bound. As you have shown experimentally, mere convergence does not result in better performance in BEAR and BEAR+SN.
> > That being said, it’s still good that you provided a convergence guarantee and the assumptions.
> >
> > P.S. I assume you meant “Figure 9” at the end of the paragraph in your reply.
> >
> > [Absorb Z into beta] Thanks for the explanation. Good to hear you’ve done experiments verifying that absorbing Z into beta does not have an impact on performance in practice.
> >
> > [Zoom in on Q-target] Thank you.

---

> > > ### Author Response · Authors · 2020-11-24
> > > **Follow up on the necessity of Theorem A.2**
> > >
> > > Thank you for providing the feedback on our answers. We are happy that most of our clarifications address your concerns well.
> > >
> > > [Theorem A.2] Thank you for pointing out this inaccurate presentation in the paper, and sorry about the misunderstanding. Indeed, by assuming that Q is bounded by Q_m, a trivial upper bound for the approximation error (2Q_m) immediately follows. However, since the final convergence bound will then depend on Q_m (Theorem B.1 [1]), the convergence bound may be less meaningful. We have updated the appendix to clarify the contribution of the proof.
> > >
> > > P.S.: Yes we meant Figure 9. Sorry about the confusion
> > >
> > > [1] Kumar, Aviral, et al. "Stabilizing off-policy q-learning via bootstrapping error reduction." Advances in Neural Information Processing Systems. 2019.

---

### Official Review · AnonReviewer4 · 2020-11-03
**Proposes novel algorithm for incorporating uncertainty estimation in offline-RL**

**Rating:** 6
**Confidence:** 4

**Review:**

Summary :
This paper considers the problem of dealing with uncertainty for static datasets in offlineRL. The authors propose a novel algorithm ‘UWAC’, uncertainty weighted actor-critic. UWAC takes a Bayesian perspective of RL, and uses Monte Carlo dropout for detecting, down-weighting OOD samples. Building on BEAR, they estimate the epistemic uncertainty as the Var(Q(s, a)), and modify the update to downweight samples with higher variance. On D4RL datasets, UWAC seems to perform competitively against model-based baselines like MOPO, model-free baselines like BEAR/CQL.

Reasons for the score:
I vote for accepting the paper, with some improvements. The paper suggests a strong improvement in the stability and performance of model-free algorithms for offlineRL. I would strongly encourage the authors to improve the writing and presentation of the algorithm in favour of clarity.


Strengths:
+ UWAC bridges the gap between the performance of model-based and model-free algorithms in offlineRL.
+ The problem is well motivated, and uses dropout variational inference to estimate the epistemic uncertainty and imposing a soft-penalizing on OOD samples.
+ Provides extensive experiments and empirical evidence on the D4RL benchmark, where it outperforms MOPO and appears to be more stable than BEAR.

Weaknesses:
- The baseline for MOPO is missing from the plots. As we are considering the stability of the algorithm as compared to other baselines (like BEAR) it would be critical to compare the same for MOPO (trained for a similar number of epochs).
- Drop in performance seems to be significantly related to exponential increase of the Q-values. Taking this into account, I would consider an implementation with some variant of clipping (gradient or reward) to be a strong missing baseline.
- Some sections were hard to follow, particularly Section 3, Section 4 could be substantially improved for clarity. Comparison among related work could probably be represented via a table of tradeoffs to better understand the strengths, nuances of the proposed algorithm.

---

> ### Author Response · Authors · 2020-11-13
> **Responses to suggestions on 1) MOPO. 2) Drop in performance.**
>
> Thank you for the positive comments that 1) our method is well-motivated, 2) achieves good performance, and 3) our experiments are extensive.
>
> We have made some minor improvements on the writing and presentation of the algorithm for clarity, and will continue to improve the presentation over the course of the rebuttal.
>
> Here are some questions we have toward your suggestions.
>
> [MOPO] MOPO is model-based. It would be hard to plot a model-based and a model-free method side-by-side since they do not train with the same training mechanism (MOPO uses extra data generated from the model). (I.e. How should we count the training epochs of the model?)
> In addition, given the current state of the problem. The performance of MOPO is on average lower than UWAC (averaged over 5 independent runs) upon convergence (at iteration 500). This already provides a fair and simple means of comparison.
>
> [Drop in performance] Gradient clipping does not solve the bootstrapping problem, and will not improve offline RL performance as shown by the non-convergence of SAC (with gradient-clipping). [1]
> The D4RL environment is already built on normalized rewards, so it is not clear to us what the benefit of reward clipping is.
> Clipping the Q function changes the optimization problem. The best constraint in our mind is a SpectralNorm. We conducted ablation studies related to SpectralNorm and showed that BEAR does not perform well even with a constrained Q function.
>
> [1] Kumar, Aviral, et al. "Stabilizing off-policy q-learning via bootstrapping error reduction." Advances in Neural Information Processing Systems. 2019.

---

### Official Review · AnonReviewer5 · 2020-11-06
**Not clear where the benefits are coming from**

**Rating:** 4
**Confidence:** 4

**Review:**

This paper proposes to use an uncertainty-weighted objective for offline RL with BEAR (Kumar et al.) that penalizes the MMD distance between the learned policy and the previous policy. The uncertainty weighted objective weights the policy improvement objective with the variance in the Q-function, where this variance primarily represents aleatoric or intrinsic uncertainty, not the epistemic or belief uncertainty. They show that their method performs reasonably better than prior methods on the D4RL datasets and show that the learned Q-values are better than BEAR.

While the work seems promising empirically, I have a number of concerns about the method:

1. The division by variance comes across as a hack -- while standard deviation could be more appropriate, I wonder why there is no need to even center the Q-function before dividing by the variance. Moreover, what does the current weighting do? Does it penalize specific state-action pairs? Can it be compared to a more standard square root inverse-counts style weighting in a tabular setting? The lack of such ablation experiments make it impossible to interpret the method.

2. Typically the notion of uncertainty in batch RL or exploration is epistemic uncertainty, where uncertainty stems from the fact that there is limited data available to the agent. The notion of uncertainty or variance considered here is not this notion. Since there are no time-consisent backups, it is unclear how epistemic uncertainty is being preserved termporally in a trajectory. The method, as it stands, seems to resemble aleatoric uncertainty more, and if this is the case, why not just use distributional critics? What is the specific beneift from dropout-based Q-functions?

3. Ablations on uncertainty: the method uses both a constraint against the behavior policy and a weighting on the policy objective. Which of these helps more and when? The Q-value plots shown in the paper are reasonable, but it is still unclear when the uncertainty still helps and why, and more understanding is needed on this part in my opinion. In fact, the Q-values plots perhaps signify my belief about 2, that their approach works because it performs "better" Q-learning similar to Agarwal et al. 2020.

Overall, I do not buy the claim that the real benefits are coming from uncertainty based Q-functions, but perhaps more so like a stronger critic function. The paper doesn't ablate over showing the benefits of uncertainty over a policy constraint or comparing to a method that actually learns Q-value estimates aware of the epistemic uncertainty making it hard to compare to these. Finally, the method does come across as adhoc. Is it possible to test the method on some Atari domains, which have some different properties than the continuous control D4RL benchmarks? This would validate if dividing by the variance is indeed transferable across domains.


------------Post Rebuttal----------------

I thank the authors for their response. I disagree about epistemic and aleatoric uncertainty. Bootstrapped DQN (Osband et al. 2016), Randomized prior functions (Osband et al. 2019), and several other works show that to get the variance of different possible Q-functions given the data, $p(Q|D)$ or differnet possible MDPs given the data, $p(M|D)$, you need backups consistent in time, i.e. the same dropout mask is to be used for both the main Q-network and the target Q-network for the backup. This is the uncertainty that we trypically need in offline RL and is also used in Theorem A.2 (for the high probability bound which is given the data), and that's why ensembles with Q-functions consistent with themelves are typically used. By merging the target values across different dropout masks, the uncertainty is not timewise consistent. While the paper that the authors point does actually do dropout masks with Q-function at each step, it is discussed in later works including Osband et al 2016, 2018 that not being consistent over time is leading to wide uncertainty estimates.  I would recommend the authors read the discussion on Posterior sampling for RL vs Optimism and Thmpson sampling for a discussion on this.

I am a bit disappointed with the rebuttal. I expected a comparison to such metods that perform timewise consistent uncertainty estimation and also to distributional RL, since the algorithm that the authors use can also be drawn similar to a set of particles of Q-functions and performing a backup using target values computed using all of the possible particles, which is essentially what, for instance, QR-DQN does in a way or even IQN does in a way. Even REM would have been fine. Without this comparison, I unfortunately cannot increase my score and I am going to retain my score.

---

> ### Author Response · Authors · 2020-11-13
> **Responses to the concerns regarding Uncertainty Estimation**
>
> Thank you for pointing out your concerns with the idea.
> Please note that the use of dropout for estimating epistemic uncertainty has already been studied in the domain of online RL (Section 5.4 of [2]). Below are our complete justifications on using dropout uncertainty estimation.
>
> [Domain Transferability]: Please note that we experimented on both MuJoCo Walkers dataset and the Adroit hand environment within the D4RL benchmarks. These datasets consist of demonstrations of different degrees of freedoms, and reward structures.
>
> [Dividing by variance]: Please note that UWAC loss is only an importance reweighing by the inverse variance. Therefore, the division by the variance in our loss does not interfere with, or introduce bias to the Q function estimation. In addition, our experiment on the lunar-lander is conducted on discrete action spaces. We also note that down-weighing with variance has the same practical effect as with std.
>
> [MC-dropout on Q function for uncertainty estimation] MC dropout on Q function has already been studied and applied to online RL to encourage exploration through Thompson Sampling (Section 5.4 of [2]). Given that the method is so simple, it certainly has some degrees of limitations. However, random prior based methods similar to dropout have been widely applied to capture uncertainty in RL [2,5,6,7,8,9]. In addition, our experiments section 5.1 empirically demonstrates that dropout uncertainty estimation can identify OOD states, which correspond to epistemic uncertainty estimation.
>
> [Epistemic v.s. Aleatoric] Aleatoric uncertainty captures noise inherent in the observations. On the other hand, epistemic uncertainty accounts for uncertainty in the model – uncertainty which can be explained away given enough data [1]. Dropout as Approximate Bayesian Inference is widely used to model the epistemic uncertainty [1,2]. Dropout uncertainty estimation has also been applied in online RL to encourage exploration in unknown states through Thompson Sampling (Section 5.4 of [2]). Our experiments section 5.1 also empirically demonstrates that dropout uncertainty estimation can identify OOD states, which correspond to epistemic uncertainty estimation.
>
> [Distributional RL] Despite the name, distributional RL and Bayesian uncertainty are quite distinct. ‘Distributional RL’ replaces a scalar estimate for the value function by a distribution that is trained to minimize a loss against the distribution of data it observes. Although both approaches might reasonably claim a ‘distributional perspective’ on RL, these two distributions have orthogonal natures and behave quite differently. Conflating one for the other can lead to arbitrarily poor decisions; it is the uncertainty in beliefs (‘epistemic’) that is captured by dropout, not the distributional noise (‘aleatoric’) as captured by the distributional RL (section 2.3 of [5]).
>
> [Ablations] We provided theoretical justifications on the design in the appendix. Empirically, the baseline performs much worse without reweighing. Different ensembling techniques have been tested by [3,4], with marginal performance benefits. Additionally, the REM proposed by Agarwal et al. 2020 requires multiple networks, and therefore is less efficient than using dropout. Also, please note that our ablation studies suggest that regularization without uncertainty re-weighting does not improve performance (see appendix A.2 and Figure 9 where we compared uncertainty estimation against a baseline with more regularized Q function). Furthermore, we conducted additional ablation studies to marginalize the effects of uncertainty weights: UWAC with dropout only (without uncertainty weighing) performs significantly worse than with uncertainty weighing (ours) on the Adroit environment (Figure 11, 12).
>
> [1] Kendall, Alex, and Yarin Gal. "What uncertainties do we need in bayesian deep learning for computer vision?." NeurIPS. 2017.
>
> [2] Gal, Yarin, and Zoubin Ghahramani. "Dropout as a bayesian approximation: Representing model uncertainty in deep learning." ICML. 2016.
>
> [3] Kumar, Aviral, et al. "Stabilizing off-policy q-learning via bootstrapping error reduction." Advances in Neural Information Processing Systems. 2019.
>
> [4] https://github.com/aviralkumar2907/BEAR
>
> [5] Osband, Ian, John Aslanides, and Albin Cassirer. "Randomized prior functions for deep reinforcement learning." Advances in Neural Information Processing Systems. 2018.
>
> [6] Fortunato, Meire, et al. "Noisy Networks For Exploration." International Conference on Learning Representations. 2018.
>
> [7] Lipton, Zachary C., et al. "Bbq-networks: Efficient exploration in deep reinforcement learning for task-oriented dialogue systems." arXiv preprint arXiv:1608.05081 (2016).
>
> [8] Tang, Yunhao, and Alp Kucukelbir. "Variational deep q network." arXiv preprint arXiv:1711.11225 (2017).
>
> [9] Touati, Ahmed, et al. "Randomized value functions via multiplicative normalizing flows." Uncertainty in Artificial Intelligence. PMLR, 2020.

---

### Author Response · Authors · 2020-11-13
**Change log**

11/23/2020:
1. Updated Appendix A.1 Theoretical analysis to emphasize that we arrive at a tighter bound on convergence.

11/14/2020:
1. Added ablation studies for UWAC w/o uncertainty weighing but with dropout (Figure 11)
2. Changed order of subplots (Figure 7,8,9,10,11) for better presentation
3. Plotted all ablations together on a same plot (Figure 12)


11/13/2020:
1. Fixed some typos
2. Updated parameter subscripts in Algorithm 1 for better clarity
3. Updated Section 4.1 for more accurate explanation of dropout uncertainty estimation
4. Added Figure 8 in the Appendix to zoom in on the Q targets
5. Added experimental details on choosing beta in the Appendix

---

### Decision · Program_Chairs · 2021-01-07
**Final Decision**

**Decision:**

Reject

**Comment:**

Being able to give confidence intervals or have a robust measure of uncertainty is very important for offline RL methods. In this work, they proposed a dropout based method to have a measure of uncertainty. The authors provide an significant empirical improvements over other baselines. Nevertheless, as it stands right now and as AnonReviewer5 have pointed out, this paper has some important shortcomings. I have noticed that the authors have updated the paper, but still the some of the important points made by AnonReviewer5 are unaddressed as it stands right now. Thus, I am suggesting to reject this paper hoping that the authors will address those issues and resubmit to a different venue.

Firstly, I agree with AnonReviewer5, it is not clear if the dropout and the variance trick used in this paper actually represents epistemic uncertainty that we would like to have for an offline RL algorithm, because the variance do not necessarily need to shrink as you train it with more data, and as opposed to supervised learning setting it is not clear what type of uncertainty the proposed dropout method will induce in the offline RL setting. It would have been nice to have some results showing how calibrated the uncertainty estimates coming from the dropout is... I would recommend the authors not to include any claims regarding the epistemic uncertainty in the camera-ready version of the paper.

Also as AnonReviewer5 pointed out, having distributional baselines and/or ensemble methods like REM or bootstrapped DQN would be a more fair comparison. So, it would be nice to see some of those baselines in a future version of this paper.